# Real-World Experience with Isavuconazole for Invasive Aspergillosis in Hematologic Patients with and Without COVID-19 in Brazil

**DOI:** 10.3390/jof11060456

**Published:** 2025-06-16

**Authors:** Larissa Simão Gandolpho, Vinicius Ponzio, Marjorie Vieira Batista, Ivan Leonardo Avelino França e Silva, Jessica Fernandes Ramos, Marcio Nucci, Arnaldo Lopes Colombo

**Affiliations:** 1Division of Infectious Diseases, Escola Paulista de Medicina-Universidade Federal de São Paulo, São Paulo 04023-900, SP, Brazil; lari.gandolpho@hotmail.com (L.S.G.); viponzio@gmail.com (V.P.); 2Hospital Nove de Julho, São Paulo 01409-002, SP, Brazil; 3A.C.Camargo Cancer Center, São Paulo 01525-001, SP, Brazil; marjorie.batista@accamargo.org.br (M.V.B.); ivan.franca@accamargo.org.br (I.L.A.F.e.S.); 4Hospital Sírio-Libanês, São Paulo 01308-050, SP, Brazil; jessicamd39@gmail.com; 5Division of Hematology, Internal Medicine Department, Universidade Federal do Rio de Janeiro, Rio de Janeiro 21941-971, RJ, Brazil; mnucci@hucff.ufrj.br; 6Grupo Oncoclínicas, Rio de Janeiro 22250-905, RJ, Brazil; 7Antimicrobial Resistance Institute of São Paulo (ARIES), São Paulo 04023-900, SP, Brazil

**Keywords:** invasive aspergillosis, isavuconazole, antifungal, opportunistic fungal infection, hematologic malignancies, COVID-19, hematopoietic stem cell transplantation

## Abstract

(1) Background: Invasive aspergillosis is a life-threatening fungal infection, particularly in patients with hematologic malignancies. Isavuconazole, a broad-spectrum triazole, has emerged as a key treatment option, but real-world data in high-risk populations from middle-income countries remain limited. (2) Methods: We conducted a multicenter, retrospective study to evaluate the clinical response rate and tolerability of isavuconazole in patients with hematologic malignancies and probable or proven invasive aspergillosis across four medical centers in Brazil. (3) Results: We enrolled 50 patients aged 18 to 82 years (64% male) with proven or probable invasive aspergillosis, diagnosed in the context of complex hematologic conditions. Among them, 60% had active or refractory malignancies, and 22% had a prior COVID-19 infection. Isavuconazole was used as a first-line therapy in 64% of cases. No patients discontinued treatment due to toxicity. The 6-week overall survival was 60%. Prior COVID-19 infection was associated with a lower survival rate (44% vs. 69% in patients without COVID-19, *p* = 0.04). (4) Conclusions: This study provides real-world evidence supporting the efficacy and tolerability of isavuconazole in a high-risk population. The findings reinforce its role as a key antifungal therapy, particularly in patients with complex underlying conditions.

## 1. Introduction

The outcomes of patients with invasive aspergillosis (IA) have improved significantly over recent decades. Improved diagnostic approaches, such as the use of fungal biomarkers (e.g., galactomannan and β-d-glucan) and enhanced imaging modalities, have contributed to earlier detection and intervention, which are critical for improving patient outcomes. In addition, the development of newer antifungal agents with expanded activity and improved safety profiles has transformed the therapeutic strategies for invasive fungal infections [1]. This progress has occurred in parallel with an expansion of high-risk populations, particularly individuals with hematologic malignancies and hematopoietic stem cell transplantation (HSCT) recipients. Despite these advances, IA remains a life-threatening infection associated with high morbidity and mortality, particularly in immunocompromised patients [1,2].

Among these antifungal agents, isavuconazole (ISAV) has emerged as a promising option for the management of IA. ISAV is a broad-spectrum triazole antifungal active against both yeasts and molds, including *Aspergillus* spp. and Mucorales [3,4]. Compared to older triazoles, ISAV exhibits several pharmacologic advantages, including predictable pharmacokinetics, a wide therapeutic window, excellent oral bioavailability, reduced nephrotoxicity and hepatotoxicity, and fewer drug–drug interactions [3,4,5,6]. These attributes are particularly important in patients with hematologic malignancies undergoing intensive chemotherapy or immunosuppression, who often experience multiple organ dysfunctions or are taking concomitant medications that significantly interact with CYP3A4 inhibitors, such as voriconazole and posaconazole [5,6].

The SECURE trial, a pivotal phase III randomized controlled trial, demonstrated that ISAV is non-inferior to voriconazole for the primary treatment of IA, while offering a superior safety and tolerability profile [7]. ISAV was associated with fewer treatment-emergent adverse events, including hepatobiliary disorders, visual disturbances, and dermatologic reactions [7,8]. Despite these promising results, the applicability of RCT findings to daily clinical practice is limited, as RCTs often exclude patients with refractory malignancies, ongoing immunosuppressive therapies, organ dysfunction, and other factors in real-world populations [9,10].

Therefore, registries and observational studies reflecting real-world experiences are of great help to assess the effectiveness and tolerability of antifungal agents in heterogeneous and clinically complex patient populations in which the rates of adverse events and outcomes may differ from those reported in RCTs [9,10,11].

Unfortunately, the access to novel antifungal agents and timely diagnosis of IA remains a major challenge in many regions of Latin America, further highlighting the need for real-world studies conducted in local populations [11]. Brazil, in particular, has seen a significant increase in the use of antifungal stewardship programs and the incorporation of newer antifungal therapies into tertiary-care hospitals; however, data describing the real-world use of ISAV in Brazilian hematologic patients remain scarce.

In this context, the present study was conducted to address this gap by summarizing the real-world experience with ISAV for the treatment of probable or proven IA in hematologic patients and HSCT recipients across four medical centers in Brazil. Our cohort included patients with relapsed or refractory hematologic malignancies, severe graft-versus-host-disease (GVHD), and recent COVID-19 infection (within the 15 days prior to IA diagnosis). All these clinical scenarios represent high-risk groups frequently excluded from clinical trials but commonly encountered in real-world hematology and transplant settings. Regarding COVID-19—an emerging risk factor for aspergillosis, candidemia, and other fungal infections—its association with hematologic malignancies may significantly increase the risk of mortality, regardless of the antifungal regimen selected as first-line therapy [12].

By characterizing clinical outcomes, treatment responses, and tolerability, we aim to contribute valuable real-world data supporting the role of ISAV in managing IA in complex and vulnerable patient populations.

## 2. Materials and Methods

This observational, retrospective, multicenter study was conducted in four Brazilian medical centers from January 2020 to April 2024. The participating centers were selected among tertiary care hospitals located in São Paulo, based on their high capacity to manage complex hematologic patients and the availability of standardized protocols for the proper prevention, diagnosis, and treatment of fungal infections in high-risk populations.

Clinical, epidemiological, and laboratory data from all consecutive episodes of IA diagnosed in adult patients (≥18 years old) with hematologic malignancies and treated with ISAV using the standard dose (as recommended by the manufacturer) were collected. Data were systematically recorded using a standardized case report form and entered into the REDCap (Research Electronic Data Capture) platform to ensure consistency and accuracy. All investigators from the four centers received previous training in data collection, including the use of the electronic clinical form and a detailed dictionary of terms developed specifically for the study. Electronic medical charts were reviewed by study coordinators to identify any inconsistencies or missing data. In case of any discrepancies, investigators were contacted to validate and, if necessary, correct the information to improve overall data quality. Particular attention was given to ensuring the completeness of microbiological, radiological, and therapeutic records.

Inclusion criteria were specifically limited to patients with hematologic malignancies and/or HSCT recipients diagnosed with IA classified by the EORTC/MSG criteria as probable or proven, to ensure diagnostic specificity [13]. Treatment response was assessed based on clinical and radiological criteria at 4, 6, and 12 weeks following IA diagnosis, when data were available. Responses were categorized by three investigators as complete response (resolution of attributable symptoms and radiological lesions, with documented clearance of infected sites), partial response (improvement of attributable symptoms and at least 25% reduction in radiological lesions, with documented clearance of infected sites), or no response (persistence or worsening of clinical symptoms, persistence of baseline radiological lesions, or an increase in fungal biomarkers) as previously described [14,15]. Putative causes of death were also evaluated by the attending physicians and were classified as IA-related or attributable to other causes.

For patients who received ISAV as initial therapy, investigators were asked to describe the rationale for its selection as the primary therapy. Similarly, for patients who received ISAV after initial treatment with another antifungal, investigators were asked to report the reasons for therapeutic switching to ISAV. The occurrence of ISAV discontinuation due to toxicity, intolerance, or clinical failure was also systematically recorded.

Continuous variables were summarized using medians and ranges, whereas categorical variables were expressed as absolute numbers and percentages. Survival analysis was performed using the Kaplan–Meier method, and comparisons between groups were made using the log-rank test, with *p*-values < 0.05 considered statistically significant. All statistical analyses were performed using SPSS Statistics software, version 28.0 (IBM Corporation, Armonk, NY, USA).

## 3. Results

During the study period, 50 cases of proven or probable IA were diagnosed. The median age of patients was 60 years (ranging from 18 to 82), and 64% were male (32 males and 18 females). As detailed in Table 1, the most common underlying hematologic malignancies were acute myeloid leukemia (*n* = 13), multiple myeloma (*n* = 11), myelodysplastic syndrome (*n* = 9), acute lymphoid leukemia (*n* = 8), non-Hodgkin lymphoma (*n* = 7), chronic lymphoid leukemia (*n* = 1), and myelofibrosis (*n* = 1). Among these patients, 30 of 50 (60%) had refractory or relapsed hematologic disease at the time of IA diagnosis.

IA was diagnosed in the setting of HSCT in 28 patients, of whom 23 had received an allogeneic HSCT and seven underwent an autologous HSCT. Among the allogeneic HSCT recipients, 18 patients developed GVHD (acute or chronic), and eight of these cases (44%) were classified as steroid-refractory GVHD.

Neutropenia within the 30 days prior to IA diagnosis was observed in 20 patients (40%), and 30 patients (60%) had received corticosteroids for indications other than GVHD management. Interestingly, 11 patients had a documented history of recent COVID-19 infection (within the 15 days prior to IA diagnosis) prior to the development of IA, including eight patients with multiple myeloma and three with non-Hodgkin lymphoma. This unusual finding highlights the increased susceptibility of hematologic patients, particularly those with lymphoproliferative disorders, to severe viral infections and subsequent opportunistic fungal complications. Finally, other relevant comorbidities included chronic obstructive pulmonary disease (COPD) in three patients and diabetes mellitus in seven patients.

The chest computed tomography (CT) results were consistent with IA in all but one patient (Table 2). The most frequent radiological features included pulmonary consolidations (*n* = 24), macronodules (*n* = 22), of which 15 cases were associated with a halo sign, and cavitary lesions in three cases. Sinus involvement was identified in 20 patients, including one case in which sinus disease was the only presenting manifestation of IA. Disseminated disease involving both the lungs and paranasal sinuses was present in 40% of cases.

Mycological diagnosis was predominantly based on galactomannan (GM) assays, which were positive in 42 patients as follows: 22 in bronchoalveolar lavage (BAL) samples, 16 in serum samples, and four in both types of biologic specimens. The median peak of the GM index was 1.74 (ranging from 0.57 to 5.46) in BAL samples and 1.65 (ranging from 0.52 to 5.20) in serum samples. In the remaining eight patients, a diagnosis was established through positive fungal cultures (*n* = 7) or histopathological confirmation and culture (*n* = 4).

ISAV was initiated as first-line therapy in 32 patients (Table 2). The most common reason for choosing ISAV as the primary treatment was concerns regarding nephrotoxicity or hepatotoxicity associated with other antifungals (*n* = 24). Additional reasons described included once-daily dosing convenience (*n* = 6) and the need to avoid drug–drug interactions (*n* = 2).

Among the 18 patients who initially received another antifungal (liposomal amphotericin B, *n* = 9; voriconazole, *n* = 9), ISAV was introduced as a second-line therapy due to the need for de-escalation (*n* = 9), subtherapeutic voriconazole levels (*n* = 5), or adverse events including hepatotoxicity (*n* = 2) and neurotoxicity (*n* = 2). Notably, no patients required the discontinuation of ISAV due to toxicity, reinforcing its favorable tolerability profile.

The clinical response and mortality rate over time of 50 episodes of IA are detailed in Table 3. At four weeks after the initiation of the ISAV therapy, a favorable clinical response, previously defined as either a complete or partial response, was observed in 36 patients (72%). A total of 13 patients (26%) died due to complications related to their underlying hematologic diseases, according to the judgement of the clinical staff. In this context, at this timepoint, only one death (2%) was considered to be directly related to IA (2%).

At six weeks, complete or partial responses were documented in 30 patients (60%). According to the judgement of the clinicians assisting the patients, the IA-related mortality remained low at 2%, although the cumulative mortality due to underlying disease increased to 38% (*n* = 19).

By twelve weeks, 27 patients (54%) had sustained a complete or partial response. The overall IA-related mortality rate remained stable at 2%, whereas the mortality from hematologic disease, comorbidities, and related complications reached 44% (*n* = 22).

As shown in Figure 1, a survival analysis using the Kaplan–Meier method revealed that prior COVID-19 infection had a significant negative impact on six-week survival. Patients without a history of COVID-19 had a six-week survival rate of 69%, compared to 44% among those who had recovered from COVID-19 prior to developing IA (*p* = 0.04, log-rank test), highlighting the additional risk conferred by previous viral infection in immunocompromised hosts.

## 4. Discussion

This multicenter, retrospective study evaluated the real-world use of ISAV for the treatment of exclusively probable or proven IA in 50 patients with hematologic malignancies across four medical centers in Brazil. Our findings reinforce the safety and effectiveness of ISAV, which was employed as a first-line therapy in 64% of cases, in a highly complex and immunocompromised population.

The clinical complexity of hematologic patients poses multiple challenges in selecting an appropriate antifungal therapy. Factors such as organ dysfunction, pharmacokinetic variability, potential drug–drug interactions, drug-related toxicities, and tolerability must be all carefully considered. Although RCTs remain the gold standard for establishing drug efficacy, safety, and approval, their strict eligibility criteria often exclude patients with advanced diseases, multiple comorbidities, significant organ impairment, or concurrent therapies [9,10]. As such, real-world studies are essential to provide complementary evidence that informs clinical decision making in real practice, in which patients frequently present overlapping vulnerabilities not represented in RCTs.

Following the SECURE trial publication, which demonstrated ISAV’s non-inferiority to voriconazole [7,8], several observational real-world studies have been published, mostly from high-resource countries, demonstrating the performance of this third-generation azole in diverse clinical scenarios and complex populations [16,17,18,19,20]. Our cohort was similarly characterized by considerable clinical complexity: 60% had relapsed or refractory hematologic malignancies, 56% underwent allogeneic HSCT, 44% of these developed steroid-refractory GVHD, and 22% experienced IA following severe COVID-19 infection.

Despite this complexity, ISAV demonstrated favorable tolerability, with no patient requiring drug discontinuation due to adverse events. This is consistent with cumulative findings from several major real-world studies involving a total of 745 patients exposed to ISAV, primarily from Europe and the USA, where discontinuation rates ranged from 2.3% to 5.3%, despite variations in patient demographics (ranging from 91 to 218 patients), heterogeneous underlying conditions, prescribing indications (prophylaxis, empirical, and targeted therapy), severity of illness, and fungal pathogens [16,17,18,19,20,21].

Interestingly, in addition to its favorable tolerability, a retrospective study involving 20 patients who switched from posaconazole to ISAV due to hepatotoxicity reported consistent improvement in liver function tests across all patients, while maintaining effective antifungal coverage [22]. This observation aligns with ISAV’s favorable safety profile, initially demonstrated in the SECURE trial and subsequently reinforced by multiple studies [7,8,16,17,18,19,20,23].

Similarly, in our study, concerns regarding hepatotoxicity and nephrotoxicity associated with other antifungal agents were the primary reasons for selecting ISAV as a first-line therapy (76%). When used as a second-line therapy, ISAV was mainly initiated due to liposomal amphotericin B de-escalation (50%), subtherapeutic voriconazole serum levels (28%), or intolerance to prior therapies (22%) (Table 2).

An important finding in our study was the negative impact of prior COVID-19 infection on patient survival outcomes. It is well established that the COVID-19 pandemic has had a profound and severe impact on immunocompromised populations, particularly patients with multiple myeloma and lymphoproliferative disorders, with reported mortality rates ranging from 30% to 45% [7,24,25,26,27]. Patients with cumulative immunosuppression, on the other hand, are particularly more susceptible to acquire bacterial and fungal superinfections [25]. COVID-19-associated pulmonary aspergillosis (CAPA), an example of these harmful complications, has been reported in approximately 10–15% of critically ill COVID-19 patients, with a high mortality rate [26,27].

In our cohort, eight of the eleven patients who developed IA after COVID-19 had multiple myeloma, while the remaining three cases were patients with non-Hodgkin lymphoma. As illustrated by the Kaplan–Meier analysis in Figure 1, patients with a prior history of COVID-19 had a significantly lower six-week survival compared to those without COVID-19 (44% vs. 69%, *p* = 0.04), as expected [27]. These findings reinforce that SARS-CoV-2 infection may further compromise host immunity and increase vulnerability in these patients, supporting its negative impact and high mortality, while highlighting the importance of aggressive surveillance and early intervention in this subgroup [24,25].

Indeed, lower response rates among patients with relapsed/refractory malignancies, persistent neutropenia, or steroid-refractory GVHD emphasize that host factors play a major role in determining IA outcomes, regardless of antifungal choice [28,29].

The overall six-week clinical response rate (complete or partial) was 60% (Table 3), which is in line with prior observational studies (ranging from 40% to 67.2%) and only slightly lower than the 72% survival rate observed in the SECURE trial at day 84 [7,8]. It is well established that clinical response rates are lower in patients with severe comorbidities, relapsed/refractory hematologic malignancies, or persistent neutropenia, emphasizing the critical role of host-related factors, particularly the status of the underlying hematologic disease, in shaping the outcomes of invasive aspergillosis, regardless of the antifungal agent used [28,29].

Our study has several limitations, including its retrospective design and relatively small sample size, as well as the data being limited to four medical centers, which restricts the statistical power of the subgroup analysis. However, a major strength of our study is the exclusive inclusion of patients with proven or probable IA documented in complex and challenging clinical scenarios who are often ineligible for RCTs. Despite the severity of the underlying conditions and comorbidities present in our patients, ISAV was associated with a favorable response rate and good tolerability.

Taken together, these findings highlight ISAV’s utility as a safe and effective antifungal agent in complex, real-world clinical settings. In addition to its broad antifungal spectrum activity, ISAV offers important pharmacologic advantages, such as its convenience of once-daily dosing, high tolerability, and low rate of treatment-limiting adverse events, which facilitate continued therapy and make it an attractive option for high-risk hematologic populations.

## 5. Conclusions

This study documents real-world experience with ISAV for the treatment of 50 patients with IA, including a high proportion (60%) of patients with relapsed or refractory hematologic malignancies who would typically be excluded from RCTs. Despite the challenging clinical scenarios posed by complex underlying hematologic conditions, combined with several comorbidities and cumulative immunosuppression, partial or complete responses were achieved in 60% of patients after six weeks of therapy. These findings are consistent with previous observational studies and contribute important additional evidence that supports ISAV’s use in real-world hematologic settings as a safe, effective, and well-tolerated therapeutic option, while also providing important insights for antifungal stewardship strategies in clinical practice.

Future prospective studies are warranted to further validate these findings and refine antifungal strategies in this vulnerable patient population.

## Figures and Tables

**Figure 1 jof-11-00456-f001:**
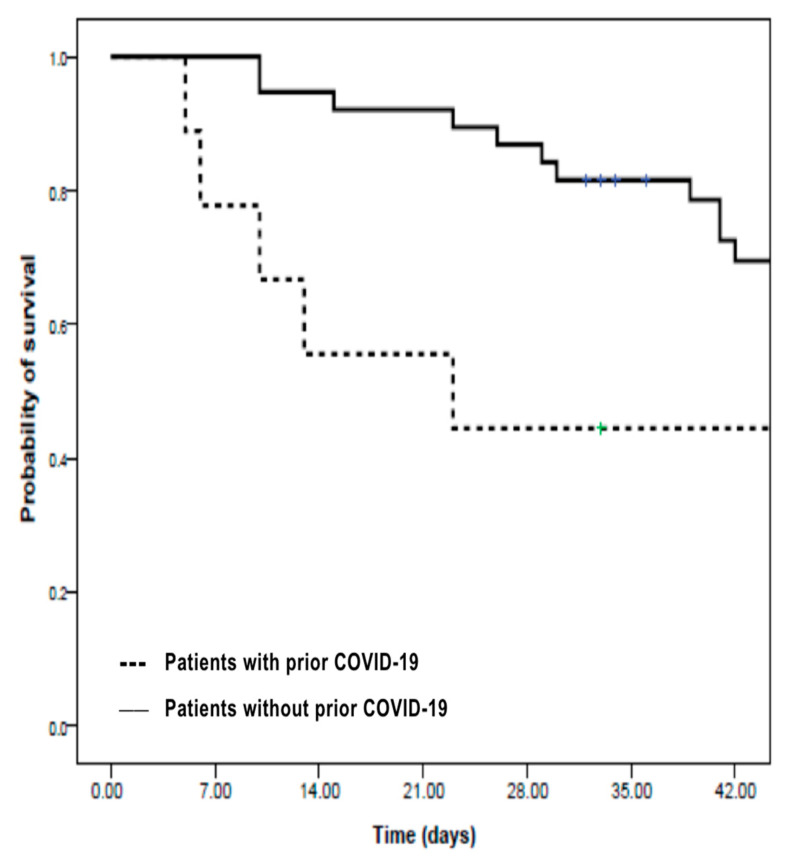
Kaplan–Meier survival curves for patients with invasive aspergillosis, stratified by prior COVID-19 infection status. Legend: Patients with a history of COVID-19 prior to IA (*n* = 11) diagnosis exhibited significantly lower 6-week survival compared to those without prior COVID-19 (*n* = 39) (44% vs. 69%, *p* = 0.04, log-rank test). This finding highlights the adverse prognostic impact of preceding SARS-CoV-2 infection in immunocompromised hosts.

**Table 1 jof-11-00456-t001:** Patients’ demographics and main baseline characteristics.

Demographics	N/Total (%)
Gender, male:female	32:18
Age, median (range)	60 (18–82)
Underlying hematologic malignancy	50/50 (100)
Acute myeloid leukemia	13/50 (26)
Acute lymphoid leukemia	8/50 (16)
Non-Hodgkin’s lymphoma	7/50 (14)
Myelodysplasia	9/50 (18)
Chronic lymphoid leukemia	01/50 (2)
Multiple myeloma	11/50 (22)
Myelofibrosis	01/50 (2)
Hematopoietic Stem Cell Transplantation	30/50 (60)
Autologous HSCT	07/30 (23)
Allogenic HSCT	23/30 (77)
Other Associated Comorbidities	
DM	07/50 (14)
Smoking	02/50 (4)
COPD	03/50 (6)
High Risk Conditions for Therapy Failure	
Acute or chronic GVHD	18/23 (78)
Steroid-refractory GVHD	8/18 (44)
Refractory/Relapsed hematologic disease	30/50 (60)
COVID-19	11/50 (22)
Multiple myeloma	8/11 (73)
Non-Hodgkin’s lymphoma	3/11 (27)

Abbreviations: GVHD = Graft-versus-host disease, HSCT = hematopoietic stem cell transplantation, COPD = Chronic obstructive pulmonary disease, DM = Diabetes mellitus.

**Table 2 jof-11-00456-t002:** Characteristics of 50 episodes of invasive aspergillosis and regimens of isavuconazole treatment.

	N/Total (%)
IA classification	
Proven	04/50 (8)
Probable	46/50 (92)
Localization of IA	
Pulmonary	49/50 (98)
Sinus (single site)	01/50 (2)
Multiple sites (pulmonary + sinus)	20/50 (40)
Administration	
Initial IV administration	47/50 (94)
Only IV administration	29/50 (58)
Rationale for Primary Therapy	
Safety profile (toxicity concerns)	24/32 (76)
Dose convenience	06/32 (18)
Drug–drug interactions	02/32 (06)
Rationale for Secondary Therapy	
Amphotericin B de-escalation	09/18 (50)
Inadequate voriconazole target concentration	05/18 (28)
Voriconazole hepatoxicity	02/18 (11)
Voriconazole neurotoxicity	02/18 (11)
Adverse events	
Increased liver function test (2- to 5-fold)	04/50 (8)
Nausea	02/50 (4)

Abbreviations: IA = Invasive aspergillosis, IV = Intravenous.

**Table 3 jof-11-00456-t003:** Clinical response and mortality rates over time of 50 patients with invasive aspergillosis.

	N (%)
4-weeks	
Complete response	24 (48)
Partial response	12 (24)
Death—Invasive aspergillosis	01 (02)
Death—Underlying disease complication	13 (26)
6-weeks	
Complete response	23 (46)
Partial response	07 (14)
Death—Invasive aspergillosis	01 (02)
Death—Underlying disease complication	19 (38)
12-weeks	
Complete response	22 (44)
Partial response	05 (10)
Death—Invasive aspergillosis	01 (02)
Death—Underlying disease complication	22 (44)

Note: Favorable response was defined as either complete or partial improvement in clinical symptoms and/or radiological findings. IA-related deaths remained stable, while mortality from underlying hematologic conditions progressively increased. Putative causes of death were evaluated by clinicians who assisted the patient (Definitions were based on [14,15]).

## Data Availability

The data presented in this study are available on request from the corresponding author. Access to the data is restricted to protect participant confidentiality.

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
