# Peer review of "Real-World Experience with Isavuconazole for Invasive Aspergillosis in Hematologic Patients with and Without COVID-19 in Brazil"

_jof, 2025, doi:10.3390/jof11060456_

Round 1
Reviewer 1 Report
The study addresses a critical gap in the literature by focusing on complex patients, including those with relapsed/refractory malignancies, hematopoietic stem cell transplantation (HSCT), and prior COVID-19 infection, who are often excluded from randomized controlled trials (RCTs). The manuscript is well-structured and contributes meaningfully to the evidence base for ISAV’s efficacy in real-world settings. The limitations are appropriately acknowledged, but the small sample size (n=50) are significant constraints that could be emphasized more. Also, study is limited to four tertiary care centers in São Paulo, which may not represent other Brazilian or Latin American settings with fewer resources. Below are comments and suggestions organized by section.
Introduction
- The introduction mentions prior COVID-19 infection in the study cohort but does not provide background on the emerging association between COVID-19 and secondary fungal infections (e.g., COVID-19-associated pulmonary aspergillosis). This context would strengthen the rationale for focusing on this subgroup.
Materials and Methods
- The study includes patients with probable or proven IA based on EORTC/MSG criteria, but it is unclear whether patients with possible IA were initially considered and then excluded or never included. If excluded describe the process for excluding possible IA cases.
Results
- The study identifies prior COVID-19 infection as a key variable, but the methods do not specify the time interval between COVID-19 and IA diagnosis. If available, report the median time (or range) between COVID-19 and IA diagnosis in the results.
- The study highlights the impact of prior COVID-19 infection on survival but does not explore other potential predictors of outcome, such as neutropenia, GVHD, or refractory malignancy status. This limits the depth of the analysis. It would be more informative to include a brief exploratory analysis of other factors associated with survival or response.
- The Kaplan-Meier survival curves are compelling, but the figure legend could be more descriptive, explaining the sample sizes for each group (COVID-19 vs. non-COVID-19).
- Typographical Errors: Correct 'COVDI-19' to 'COVID-19' (line 29), replace 'revived' with 'received' (line 100), and correct the formatting of 'Hematopoietic Stem Cell Transplantation (HSCT)' in Table 1."
The study addresses a critical gap in the literature by focusing on complex patients, including those with relapsed/refractory malignancies, hematopoietic stem cell transplantation (HSCT), and prior COVID-19 infection, who are often excluded from randomized controlled trials (RCTs). The manuscript is well-structured and contributes meaningfully to the evidence base for ISAV’s efficacy in real-world settings. The limitations are appropriately acknowledged, but the small sample size (n=50) are significant constraints that could be emphasized more. Also, study is limited to four tertiary care centers in São Paulo, which may not represent other Brazilian or Latin American settings with fewer resources. Below are comments and suggestions organized by section.
Introduction
- The introduction mentions prior COVID-19 infection in the study cohort but does not provide background on the emerging association between COVID-19 and secondary fungal infections (e.g., COVID-19-associated pulmonary aspergillosis). This context would strengthen the rationale for focusing on this subgroup.
Materials and Methods
- The study includes patients with probable or proven IA based on EORTC/MSG criteria, but it is unclear whether patients with possible IA were initially considered and then excluded or never included. If excluded describe the process for excluding possible IA cases.
Results
- The study identifies prior COVID-19 infection as a key variable, but the methods do not specify the time interval between COVID-19 and IA diagnosis. If available, report the median time (or range) between COVID-19 and IA diagnosis in the results.
- The study highlights the impact of prior COVID-19 infection on survival but does not explore other potential predictors of outcome, such as neutropenia, GVHD, or refractory malignancy status. This limits the depth of the analysis. It would be more informative to include a brief exploratory analysis of other factors associated with survival or response.
- The Kaplan-Meier survival curves are compelling, but the figure legend could be more descriptive, explaining the sample sizes for each group (COVID-19 vs. non-COVID-19).
- Typographical Errors: Correct 'COVDI-19' to 'COVID-19' (line 29), replace 'revived' with 'received' (line 100), and correct the formatting of 'Hematopoietic Stem Cell Transplantation (HSCT)' in Table 1."
Author Response
Comment 1: The introduction mentions prior COVID-19 infection in the study cohort but does not provide background on the emerging association between COVID-19 and secondary fungal infections (e.g., COVID-19-associated pulmonary aspergillosis). This context would strengthen the rationale for focusing on this subgroup.
Response 1: We appreciate the reviewer’s valuable comment. We have now included background information on the emerging association between COVID-19 and secondary fungal infections, including COVID-19-associated pulmonary aspergillosis, to better contextualize the focus on this subgroup. Please see lines 83–88.
Comment 2: The study includes patients with probable or proven IA based on EORTC/MSG criteria, but it is unclear whether patients with possible IA were initially considered and then excluded or never included. If excluded describe the process for excluding possible IA cases.
Response 2: Thank you for this observation. We agree that our initial wording may have lacked clarity. We have now improved the description of the methodology to clarify that patients classified as having possible IA were not included in the study. Please see lines 113–115.
Comment 3: The study identifies prior COVID-19 infection as a key variable, but the methods do not specify the time interval between COVID-19 and IA diagnosis. If available, report the median time (or range) between COVID-19 and IA diagnosis in the results.
Response 3: We thank the reviewer for highlighting this important detail. The upper limit of time between COVID-19 infection and IA diagnosis considered for accepting the coexistence of both conditions has now been added to the Results section. Please refer to lines 158–159.
Comment 4: The study highlights the impact of prior COVID-19 infection on survival but does not explore other potential predictors of outcome, such as neutropenia, GVHD, or refractory malignancy status. This limits the depth of the analysis. It would be more informative to include a brief exploratory analysis of other factors associated with survival or response.
Response 4: We fully agree with the reviewer that other factors such as neutropenia, GVHD, and refractory malignancies can also significantly affect outcomes in patients with invasive aspergillosis. However, these aspects have been addressed in the Results section (lines 156–164) and further discussed in the Discussion section (lines 293–295).
Comment 5: The Kaplan-Meier survival curves are compelling, but the figure legend could be more descriptive, explaining the sample sizes for each group (COVID-19 vs. non-COVID-19).
Response 5: As recommended, we have revised the figure legend to provide a clearer description, including the sample sizes for the COVID-19 and non-COVID-19 groups. Please see lines 321–324.
Comment 6: Typographical Errors: Correct 'COVDI-19' to 'COVID-19' (line 29), replace 'revived' with 'received' (line 100), and correct the formatting of 'Hematopoietic Stem Cell Transplantation (HSCT)' in Table 1."
Response 6: Thank you for pointing out these typographical errors. All mentioned corrections have been made accordingly.
Reviewer 2 Report
This paper describes a cohort of immune compromised patients that received ISAV for IA. The descriptions of the cohort and the response to treatment are interesting and I appreciate the challenge of studying a patient population that would not be in a RCT. The discussion is overly broad for a retrospective study with a relatively small sample size and the fact that all patients received ISAV.
Abstract
Line 29: COVID typo
Introduction
Lines 38-48: It seems like this would need more than 1 reference.
Materials and Methods
Clear inclusion/exclusion criteria should be added. Were any cases removed from analysis?
Results
Lines 145-147: Who are the other 40%?
Table 1: the label n/total (%) starts with underlying hematologic malignancy and should be moved below demographics. The underlying hematologic malignancy and comorbidities groups are missing the /total part. I don’t understand why there are different denominators in HSCT (the GVHD groups have 23 or 18 patients vs 30 in the HSCT groups). Autologous and allogenic HSCT numbers do not match the text.
Lines 154-155: What was the timeline on previous covid infection? Ever? Or within a few months of the IA?
Table 2: Missing some of the /total numbers.
Figure 1: All but one of the deaths was attributed to the underlying disease, not IA. I do not understand how ISAV and IA fit into this picture. It says more about a possible relationship between Covid and death due to the underlying disease.
Line 183: 2 patients are unaccounted for in the initial other antifungal group.
What dose of ISAV was used? Length of treatment?
Discussion
I don’t think Table 4 is appropriate for the manuscript, particularly as the cited studies are not for IA specifically. The summary in the text is excellent and can be referenced.
Lines 269-278: Did the patients in this study have hepatic and renal monitoring performed? Was everything normal?
285-287: I understand the covid connection a little better, but you’re relating covid to the hematological deaths in this study, which doesn’t say anything about the use of ISAV in this cohort.
Lines 297-299: Is this true for your cohort as well?
Author Response
Comment 1: Line 29: COVID typo.
Response 1: Thank you for pointing out this typographical error. The correction has been made accordingly.
Comment 2: Lines 38-48: It seems like this would need more than 1 reference.
Response 2: Thank you for this observation. Additional references have been added as suggested to strengthen the contextual background.
Comment 3: Clear inclusion/exclusion criteria should be added. Were any cases removed from analysis?
Response 3: We agree with the reviewer that additional clarity was needed. We have now improved the description of the methodology to explicitly detail the inclusion criteria. No cases meeting the eligibility criteria were excluded from analysis.
Comment 4: Lines 145-147: Who are the other 40%?
Response 4: Thank you for this question. Lines 147–150 describe the underlying hematologic malignancies in our cohort of 50 patients. As stated in lines 150–151, 60% had relapsed or refractory disease. Therefore, the remaining 40% had hematologic malignancies that were considered responsive at the time of IA diagnosis.
Comment 5: Table 1: The underlying hematologic malignancy and comorbidities groups are missing the /total part.
Response 5: Thank you for the comment. The proportions (/total) have been added as requested in Table 1.
Comment 6: Lines 154-155: What was the timeline on previous covid infection? Ever? Or within a few months of the IA?
Response 6: We appreciate the reviewer’s suggestion. The median time between COVID-19 infection and IA diagnosis has now been reported in the Results section (lines 158–159), clarifying that only cases within 15 days prior to IA diagnosis were considered.
Comment 7: Table 2: Missing some of the /total numbers.
Response 7: The mentioned corrections have been made accordingly, as well as to the other tables for consistency.
Comment 8: Line 183: 2 patients are unaccounted for in the initial other antifungal group.
Response 8: Thank you for this observation. We have corrected the patient count in the “other antifungal” group in line 189 to ensure consistency with the overall numbers.
Comment 9: What dose of ISAV was used? Length of treatment?
Response 9: As requested, we specified in line 103 that all patients received the standard recommended dose of isavuconazole, according to the manufacturer’s labeling. The length of treatment varied depending on the resolution of the underlying hematologic condition and was not used as a surrogate marker of efficacy once most patients demanded prolonged antifungal secondary prophylaxis due to the refractoriness and complications of the underlying hematologic conditions.
Comment 10: I don’t think Table 4 is appropriate for the manuscript, particularly as the cited studies are not for IA specifically. The summary in the text is excellent and can be referenced.
Response 10: We appreciate the reviewer’s thoughtful feedback and fully agree. Table 4 has been removed from the manuscript, as the summary provided in the text is indeed sufficient.
Comment 11: Lines 269-278: Did the patients in this study have hepatic and renal monitoring performed? Was everything normal?
Response 11: As mentioned in the Methods (lines 128–129) and Results (lines 191–192), all patients were systematically monitored for signs of toxicity or intolerance. Despite the presence of underlying conditions in some individuals (e.g., GVHD, pre-existing hepatic dysfunction), no patient required discontinuation of isavuconazole due to adverse events.
Comment 12: 285-287: I understand the covid connection a little better, but you’re relating covid to the hematological deaths in this study, which doesn’t say anything about the use of ISAV in this cohort.
Response 12: With all respect to the reviewer’s concern, our intention throughout the manuscript was to present real-world outcomes of ISAV use in complex patients often underrepresented in clinical trials. The inclusion of patients with prior COVID-19, refractory hematologic malignancies, or GVHD reflects real-world clinical challenges that may attenuate treatment outcomes regardless of the antifungal used. Despite these factors, ISAV demonstrated satisfactory efficacy and tolerability in our cohort.
Comment 13: Lines 297-299: Is this true for your cohort as well?
Response 13: As stated in lines 191–192, no patients in our cohort required ISAV discontinuation due to adverse effects, supporting its favorable safety profile in this real-world setting.
Reviewer 3 Report
I congratulate the authors for this interesting and useful study, in which they present the real-life experience of using isavuconazole in patients with IA and complex underlying diseases. This information is of great interest to the field of study. Although the work has limitations, which the authors openly acknowledge, the scientific rigor of the work is evident. I have only minor observations:
Since the study also provides clinical and epidemiological data on IA, it would be advisable to include the species in cases where the etiologic agent has been identified.
Table 1 GVDH and HSCT need to be defined (remove HSCT from the table subtitle).
Tables 2 and 3. Express the percentage without a zero, i.e., 8, not 08. I suggest placing the percentage in a separate column.
Figure 1. The title and explanation of Figure 1 should be included at the bottom of the figure.
References in JoF format.
Line 51 spp. is not italicized.
Lines 133-135 should include the approval date.
Author Response
Comment 1: Table 1 GVDH and HSCT need to be defined (remove HSCT from the table subtitle).
Response 1: We thank the reviewer for the suggestion. The abbreviations have been properly defined, and "HSCT" was removed from the table subtitle as recommended.
Comment 2: Tables 2 and 3. Express the percentage without a zero, i.e., 8, not 08. I suggest placing the percentage in a separate column.
Response 2: Thank you once again for the helpful feedback. We have reformatted the tables to remove the leading zeroes and placed the percentages in a separate column for better readability.
Comment 3: Figure 1. The title and explanation of Figure 1 should be included at the bottom of the figure.
Response 3: The figure legend has been repositioned below the figure, as suggested.
Comment 4: Lines 133-135 should include the approval date.
Response 4: The approval date of the study protocol has now been included in the specified lines, as requested.
Round 2
Reviewer 2 Report
My questions were all addressed. Thank you to the authors!
N/A